# AEDG Peptide (Epitalon) Stimulates Gene Expression and Protein Synthesis during Neurogenesis: Possible Epigenetic Mechanism

**DOI:** 10.3390/molecules25030609

**Published:** 2020-01-30

**Authors:** Vladimir Khavinson, Francesca Diomede, Ekaterina Mironova, Natalia Linkova, Svetlana Trofimova, Oriana Trubiani, Sergio Caputi, Bruna Sinjari

**Affiliations:** 1Saint Petersburg Institute of Bioregulation and Gerontology, Dynamo Ave., 3, 197110 St. Petersburg, Russia; ibg@gerontology.ru (V.K.); katrine1994@mail.ru (E.M.); dr.s.trofimova@gmail.com (S.T.); 2Pavlov Institute of Physiology Russian Academy of Sciences, Makarova Emb., 6, 199034 St. Petersburg, Russia; 3Department of Medical, Oral and Biotechnological Sciences, University “G. d’Annunzio” Chieti-Pescara, Via dei Vestini, 31, 66100 Chieti, Italy; francesca.diomede@unich.it (F.D.); trubiani@unich.it (O.T.); info@unich.it (S.C.); sinjaribruna@gmail.com (B.S.); 4Academy of postgraduate education under FSBU FSCC of FMBA of Russia, Volokolamskaya r., 91, 125371 Moscow, Russia

**Keywords:** AEDG peptide (Ala-Glu-Asp-Gly, Epitalon), human gingival mesenchymal stem cells, neurogenic differentiation, histones, epigenetic

## Abstract

It was shown that AEDG peptide (Ala-Glu-Asp-Gly, Epitalon) regulates the function of the pineal gland, the retina, and the brain. AEDG peptide increases longevity in animals and decreases experimental cancerogenesis. AEDG peptide induces neuronal cell differentiation in retinal and human periodontal ligament stem cells. The aim of the study was to investigate the influence of AEDG peptide on neurogenic differentiation gene expression and protein synthesis in human gingival mesenchymal stem cells, and to suggest the basis for the epigenetic mechanism of this process. AEDG peptide increased the synthesis of neurogenic differentiation markers: Nestin, GAP43, β Tubulin III, Doublecortin in hGMSCs. AEDG peptide increased Nestin, GAP43, β Tubulin III and Doublecortin mRNA expression by 1.6–1.8 times in hGMSCs. Molecular modelling method showed, that AEDG peptide preferably binds with H1/6 and H1/3 histones in His-Pro-Ser-Tyr-Met-Ala-His-Pro-Ala-Arg-Lys and Tyr-Arg-Lys-Thr-Gln sites, which interact with DNA. These results correspond to previous experimental data. AEDG peptide and histones H1/3, H1/6 binding may be one of the mechanisms which provides an increase of Nestin, GAP43, β Tubulin III, and Doublecortin neuronal differentiation gene transcription. AEDG peptide can epigenetically regulate neuronal differentiation gene expression and protein synthesis in human stem cells.

## 1. Introduction

The main regulatory systems of multicellular organisms, the nervous, endocrine, and immune systems, have the same mechanisms of peptide regulation at autocrine, endocrine, neurocrine, and paracrine levels. Peptides can be divided into two groups: short peptides (oligopeptides having up to 10 amino acids) and long peptides (from several dozens to hundreds of amino acids, e.g., insulin and growth factors). Oligopeptides are especially relevant nowadays and different studies have shown their effect on cell senescence and ageing. It was shown that short peptides play an important role in the transmission of biological information, modulation of transcription, and restoration of genetically conditioned alterations occurring with age [1,2,3,4,5]. Some peptide bioregulators with geroprotective activity were created and studied at St. Petersburg Institute of Bioregulation and Gerontology (Russia) more than 40 years ago [6,7].

The correction of pathological consequences of cerebral microvascular network age-related devastation is a pressing problem in physiology and medicine. On the other hand, lifespan prolongation in countries with a well-developed economy is associated with an increase in patient numbers with different forms of dementia: Alzheimer, Parkinson and Huntington diseases, encephalopathy and angiopathies. Methods for the activation of therapeutic angiogenesis in brain aging are being developed now [8,9]. Moreover, neurodegenerative diseases such as Alzheimer’s and Parkinson’s are devastating due to the fact that neurons cannot be replaced in the diseased areas of the brain. The generation of new neurons from stem cells, however, has the potential to become a viable treatment option for patients with brain disorders [10,11].

Mesenchymal stem cells (MSCs) are able to differentiate into skeletal and chondrogenic tissues, and also in neurons and glial cells [12,13]. Oral tissues are considered an easily accessible source of MSCs with no ethical issues. In the oral cavity, six different adult human dental stem cells have been described. Orally-derived MSCs, particularly human gingival mesenchymal stem cells (hGMSCs), are widely used in stem cell therapy for regenerative purposes in combination with or without different biomaterials to enhance the osteogenic process [14,15,16]. HGMSCs can be considered an easily accessible stem cell population, migrated from the neural crest [17,18]. They are particularly apt as an in vitro study model in regenerative medicine and in systemic diseases [19,20]. The latter, due to their origin from the neural crest, seem more suitable for differentiating toward the neuronal lineage [21]. In addition, the preconditioning of MSCs can improve their beneficial effects. 

Short peptides (di-, tri-, and tetrapeptides) are signaling molecules capable of interacting with DNA and histone proteins, acting as regulatory factors [2,5,22,23,24]. Not all peptides can interact with DNA. In previous investigations we systematically analyzed the ability of dipeptides (all possible combinations of the 20 standard amino acids) to bind all possible combinations of tetra-nucleotides in the central part of double strength DNA (ds DNA) by molecular modelling methods. We identified only 57 low-energy dipeptide complexes with peptide-dsDNA possessing high selectivity for dsDNA binding. The analysis of the dsDNA complexes with dipeptides with free and blocked N- and C-terminus showed that selective peptide binding to dsDNA can increase dramatically with peptide length. For example, KE dipeptide (Lys-Glu, Vilon), which have immonoprotective and other biological activity in vitro and in vivo experiments, can selective bind TCGA DNA sequence [25]. It was previously shown, that AEDG (Ala-Glu-Asp-Gly, Epitalon), EDR (Glu-Asp-Arg, Pinealon), AEDL (Ala-Glu-Asp-Leu, Bronchogen), KEDW (Lys-Glu-Asp-Trp-NH_2_, Pancragen) peptides can specifically bind with FITC-labeled H1, H2b, H3 and H4 histones [26]. There is not sufficient data about the systematically research of all di-, tri- and tetrapeptides and all histones protein interaction. But we can suppose, that not all short peptides can interact with histone protein, because some short peptides (for example AEDG, EDR, AEDL, KEDW) can have different biological activity and, may be, different molecular target points in the cell. 

It was shown that short peptides regulate functional activity, proliferation, apoptosis and differentiation of human, animal and plant cells. 

In our previous investigation it was shown, that AEDG and other short peptides promote neuronal differentiation of human periodontal ligaments stem cells (hPDLSCs). Western Blot analysis and immunofluorescent confocal microscopy demonstrated that the compound of AEDG, KED, KE, and AED peptides increased grow-associated protein 43 (GAP43) and nestin (neurofilament protein) synthesis in hPDLSCs culture [27]. Moreover, earlier it was shown that AEDG peptide involved in the regulation of proliferation and differentiation processes in various types of cells and tissues demonstrated geroprotective effects in animal models. AEDG peptide prolonged the life of animals increasing telomere length [3,28]. AEDG peptide increased melatonin synthesis in the pineal gland during ageing [29,30]. AEDG peptide induced retinal cell differentiation [31,32] and normalized renal function in a pathology model in rats [33]. We suggested that the possible mechanism of AEDG peptide regulation of cells functions is connected with site-specific interactions of peptides with H1, H2b, H3, and H4 histone tails in chromatin [26].

The aim of the study is to investigate the influence of AEDG peptide on neurogenic differentiation gene expression and protein synthesis (nestin, β tubulin III, GAP43, doublecortin) in hGMSCs and to suggest the basis for the epigenetic mechanism of this process.

## 2. Results

### 2.1. Cell Characterization

The expression of different surface molecules CD13, CD29, CD44, CD73, CD90 and CD105 were analyzed in hGMSCs, with the cells being negative for the subsequent markers CD14, CD34, and CD45 (Table 1). 

To evaluate the mesenchymal feature, the cells were induced to adipogenic and osteogenic commitment. To evaluate the ability to differentiate into osteogenic lineage, hGMSCs were stained with Alizarin Red S to observe calcium depositions (Figure 1B). The analysis of transcripts RUNX-2 and ALP confirmed the ability of both cell types to differentiate (Figure 1A). RUNX-2 and ALP mRNA expression in differentiated cells was 1.6 and 1.8 times higher than that in undifferentiated cells. To evaluate the adipogenic differentiation of hGMSCs, the cells were stained with Oil Red O and observed by light inverted microscopy. The cells showed several intracellular lipid vacuoles at cytoplasmic level (Figure 1D). These data were validated by the upregulation of FABP4 and PPARγ (Figure 1C). FABP4 and PPARγ mRNA expression in differentiated cells was 3.0 and 2.2 times higher than that in undifferentiated cells.

### 2.2. Immunofluorescence Analysis

HGMSCs were treated with AEDG peptide in concentration 0.01 μg/mL for 1 week. After the treatment period, the cells were observed by confocal microscopy to evaluate the modulation in marker expression related to neurogenic differentiation. HGMSCs treated with AEDG peptide showed upregulation of all the studied markers: Nestin, GAP43, β-tubulin III and Doublecortin (Figure 2).

### 2.3. Gene Expression

Transcript levels of Nestin, GAP43, β-tubulin III and Doublecortin were analyzed by RT-PCR. Neurogenic-related genes were upregulated in hGMSCs treated with AEDG peptide for one week (Figure 3). AEDG peptide increased Nestin, GAP43, β-tubulin III and Doublecortin mRNA expression by 1.7, 1.6, 1.8, and 1.7 times in cell culture in comparison with untreated cells.

### 2.4. Histone-Peptide Interaction Analysis

The calculated low-energy conformation of the AEDG peptide is shown in Figure 4. At pH 7, the total charge of the peptide molecule is −2. The AEDG peptide forms four4 intramolecular hydrogen bonds, dotted in Figure 5. The energy of this low-energy conformation was −294.43 kcal/mol. Alanine (A)—a hydrophobic amino acid, has a nonpolar radical. Glutamic acid (E) and aspartic acid (D) are hydrophilic amino acids with negatively charged polar radicals. Glycine (G) has a non-polar radical. The hydrophobicity index of the AEDG peptide was calculated according to the Kite-Dullit table as the sum of the hydrophobicity index of each amino acid residue included in the peptide. A high hydrophobicity index indicates a high degree of hydrophobicity of the molecule. The hydrophobicity index of the AEDG peptide is −8.5, which indicates the hydrophilicity of the molecule.

For the interaction of the AEDG peptide with histones H1/1, H1/3, H1/6, H2b, H3, H4, the first 50 docking solutions were analyzed. In the interaction of the AEDG peptide with histone H1/1, 13 solutions were found at site 2, 4 solutions at site 1, 7 solutions at site 3, and 3 solutions at site 4. The AEDG peptide binds to histone H1/1 according to the sequence Ile-Thr-Leu-Lys-Glu-Arg-Thr-Gly-Val-Ala-Lys-Lys with a minimum energy of −27.29 kcal/mol. During the interaction of the AEDG peptide with histone H1/3, 11 solutions were found at site 1, 1 solution at site 2, 3 solutions at site 3, 8 solutions at site 4, and 14 solutions at site 5. The main binding site of the AEDG peptide is located at site 5, which interacts with DNA. The AEDG peptide binds to histone H1/3 according to the sequence His-Pro-Ser-Tyr-Met-Ala-His-Pro-Ala-Arg-Lys with an energy of −56.49 kcal/mol. During the interaction of the AEDG peptide with histone H1/6, 5 solutions were found at site 1, 4 solutions at site 2, 3 solutions at site 3, and 12 solutions at site 5. The main binding site of the AEDG peptide is located at site 5, which interacts with DNA. Figure 5A,B shows one of the docking solutions. The AEDG peptide binds to histone H1/6 in the sequence Tyr-Arg-Lys-Thr-Gln with an energy of −64.51 kcal/mol. This energy value is the minimum (most probable) of all the models constructed in our work. During the interaction of the AEDG peptide with histone H2b, 12 solutions were found at site 3, 5 solutions at site 4, and 1 solution at site 6. The main binding site of the AEDG peptide is located at the N-end of the histone and in the second alpha helix, which is located near the N-end (site 3). The AEDG peptide binds to histone H2b at the N-terminal domain according to the sequence Val-Glu-Thr-Ser-Asn-Ser-Asn with an energy of −23.10 kcal/mol. During the interaction of the AEDG peptide with histone H3, 11 solutions were found at site 2, 8 solutions at site 3, 3 solutions at sites 5, 6 and 7, 4 solutions at site 8, and 1 solution at site 9. The main binding site of the AEDG peptide is located at the N-terminus of histone. The AEDG peptide binds to histone H3 at the N-terminal domain according to the sequence Lys-Ser-Thr-Lys-Arg-Lys with an energy of −27.44 kcal/mol. Thus, the AEDG peptide binds to the N-terminal portion (“tail”) of histone H3. As part of nucleosomes, “tails” are exposed outside the core particle and are often targeted by enzymes that modify the structure of histones. During the interaction of the AEDG peptide with histone H4, 14 docking solutions were found at site 1, 5 solutions at site 2, 16 solutions at site 3, and 7 solutions at site 4. The AEDG peptide binds to histone H4 at the N-terminal domain according to the sequence Lys-Arg-His-Val-Leu-Arg-Asp-Asn with a minimum energy of −27.41 kcal/mol.

## 3. Discussion

Teeth and their surrounding tissues represent an important area of study in regenerative medicine, because of their unique and complex developmental origin and the invasiveness of cell collection. Six different types of MSCs have been characterized in the oral cavity. One of them is human hGMSCs, which has been widely studied by our group [34,35,36,37]. In the present study both types of these cells, collected from healthy volunteers (five men in the age range from 20 to 30), were first studied for their stemness characteristics. This evaluation was performed through an assessment of the expression of different surface molecules such as: CD13, CD29, CD44, CD73, CD90, CD105, and markers CD14, CD34, and CD45 [38]. The first group of stemness markers were highly expressed in both types of cells, while the others like CD14, CD34, CD45 were not expressed at all. On the other hand, to evaluate the mesenchymal feature, the cells were induced to adipogenic and osteogenic commitment. In fact, as shown in Figure 1, the results of the present study demonstrated that these kind of stem cells have the ability to differentiate in osteogenic and adipogenic lineage. Moreover, these results of the multi-lineage differentiation capacity were validated by the upregulation of the FABP4 known as intracellular lipid chaperones, which regulate lipid trafficking and responses in cells [39,40], as well as PPARγ critically involved in the regulation of a large number of genes, which regulate energy homeostasis, glucose triglyceride and lipoprotein metabolism, de novo lipogenesis, fatty acid uptake, oxidation, storage and export, cell proliferation, inflammation, and vascular tissue function [41]. 

Immunofluorescence and RT-PCR results have shown the ability of hGMSCs, treated with AEDG peptide, to differentiate in neurogenic commitment. Human GMSCs treated with AEDG peptide showed an upregulation of all the genes in question and proteins: Nestin, GAP43, β Tubulin III and Doublecortin (Figure 3 and Figure 4). These results were in accordance with other results published previously by the present groups, where the AEDG peptide and the mixture of AEDG, KED, KED and AED peptides increased expression GAP43 and Nestin genes and proteins in hPDLSCs [27]. Thus, the reason for many biological activities and geroprotective effects of the AEDG peptide could lie in the influence of this peptide on gene expression and protein synthesis during neurogenesis. How can the AEDG peptide regulate gene expression?

Previously, the regulatory regions of DNA were considered as potential targets of epigenetic action for the AEDG peptide in the cell [2]. It is assumed that the site-specific interaction of the AEDG peptide with DNA leads to a change in the nature of gene expression. The article provides evidence that the epigenetic regulation of gene expression by the AEDG peptide can also be realized through interaction with core and linker histones. We have created models for the interaction of the AEDG peptide with histones H1/1, H1/3, H1/6, H2b, H3, and H4. In contrast to the linker histones H1/1, H1/3, H1/6, the core histones H4, H3, H2b did not reveal any homologous sites for the binding of the AEDG peptide [26]. In our study, the AEDG peptide was most likely to interact with the linker histones H1/6, as requiring the lowest binding energy (sequence Tyr-Arg-Lys-Thr-Gln, binding energy −64.51 kcal/mol) and H1/3 (sequence His-Pro-Ser-Tyr-Met-Ala-His-Pro-Ala-Arg-Lys, binding energy −56.49 kcal/mol). This is in accordance with the experimental data in the article [26]. Moreover, the AEDG peptide is less likely to form complexes with the linker histone H1/1 and core histones H4, H3, H2b. It was found that the AEDG peptide most preferably binds to histones H1/6 and H1/3 at sites that interact with DNA. We hypothesize that peptides compete with the histones at these sites of the DNA molecule. In this regard, it has been suggested that the binding of the peptide to histones can serve as an additional mechanism which increases the probability of transcription of the genes encoding proteins involved in neuronal stem cell differentiation, such as Nestin, GAP43, β Tubulin III and Doublecortin (Figure 6). Thus, the AEDG peptide epigenetically regulates gene expression and protein synthesis, the markers of neuronal differentiation of human stem cells.

## 4. Materials and Methods

### 4.1. Cell Cultures

The study was approved by the Ethical Committee of the University “G. d’Annunzio”, Chieti and Pescara (PI: Prof. Trubiani Oriana; N266/2014). To evaluate the mesenchymal features of hGMSCs, cytofluorimetric detection and mesengenic differentiation have been performed. Cytofluorimetric analysis was assayed as previously described [40]. Expression of Oct3/4, Sox-2, SSEA-4, CD14, CD29, CD34, CD44, CD45, CD73, CD90, and CD105 was evaluated on hGMSCs. The analysis was performed using FACStarPLUS flow cytometry system and FlowJo™ software (TreeStar, Ashland, OR, USA).

To assess their ability to differentiate into osteogenic and adipogenic commitment, hGMSCs were maintained under osteogenic and adipogenic conditions for 21 and 28 days respectively, as previously described [42]. To evaluate the formation of mineralized precipitates and lipid vacuoles, after the differentiation period, alizarin red and adipo oil red staining were performed on undifferentiated and differentiated hGMSCs. Inverted light microscopy Leica DMIL (Leica Microsystem, Milan, Italy) was used for sample observations. To validate their ability to differentiate into osteogenic and adipogenic lineages, the expression of RUNX-2, ALP, FABP4, and PPARγ were evaluated using real time polymerase chain reaction (RT-PCR) as reported [43]. Commercially available TaqMan Gene Expression Assays (RUNX-2 Hs00231692_m1; ALP Hs01029144_m1; FABP4 Hs01086177_m1; PPARγ Hs01115513_m1) and Taq-Man Universal PCR Master Mix (Applied Biosystems, Foster City, CA, USA) were used according to standard protocols. Beta-2 microglobulin (B2M Hs99999907_m1) (Applied Biosystems) was used for template normalization. RT-PCR was performed in three independent experiments, and duplicate determinations were carried out for each sample.

### 4.2. Experimental Design

HGMSCs were cultured until the thirds passage and then divided into 2 groups: first—control group (without adding peptide), and second—the ones treated with AEDG peptide. The peptide was diluted in a sodium phosphate-buffered saline (PBS) buffer in concentration 0.01 μg/mL. The peptide was added to the cell medium and replaced every 3 days [42]. The cells were placed at 37 °C in a humidified 5% CO_2_ incubator. On the 7th day of peptide treatment, the cell cultures were analyzed by immunofluorescence and RT-PCR for the neurogenic differentiation marker expression.

### 4.3. Immunofluorescence Analysis

The cells were fixed for 30 min at room temperature with 4% of paraformaldehyde in 0.1 M PBS, 40 pH 7.4, and permeabilized with 0.1% of Triton1-X100 in PBS for 10 min, followed by blocking with 5% skimmed milk in PBS for 30 min. The samples were incubated with mouse primary monoclonal antibody, anti-Nestin 1:200 (Santa Cruz Biotechnology, Inc., Dallas, TX, USA), anti-GAP43 (1:500; Sigma Aldrich, Milan, Italy), anti-beta Tubulin III (1:250; Santa Cruz Biotechnology) and anti-Doublecortin (1:200; Abcam, DBA, Milan, Italy) as the primary antibody and anti-mouse Alexa Fluor 568 probe (Molecular Probes) as the secondary antibody. All samples were incubated with Alexa Fluor 488 phalloidin green fluorescence conjugate (1:200), as a marker of the cytoskeleton actin and with TO-PRO staining to stain the nuclei [42]. The samples were observed using a Zeiss LSM800 META confocal (Zeiss, Jena, Germany) connected to an inverted Zeiss Axiovert 200 microscope equipped with a Plan Neofluar oil-immersion objective (40×/1.3 NA). The images were collected using an argon laser beam with excitation lines at 488 nm and a helium–neon source at 543 and 633 nm. 

### 4.4. Real Time PCR

The neurogenic markers were evaluated by RT-PCR. To this end, the total RNA was isolated using the Total RNA Purification Kit (NorgenBiotek Corp., Ontario, CA, USA) according to the manufacturer’s instructions. M-MLV Reverse Transcriptase reagents (Applied Biosystems) were used to generate cDNA. RT-PCR was carried out with the Mastercycler ep realplex real-time PCR system (Eppendorf, Hamburg, Germany). Commercially available TaqMan Gene Expression Assays (Nestin Hs04187831_g1; GAP43 Hs00967138_m1; β Tubulin III Hs00801390_s1; Doublecortin Hs00167057_m1) and Taq- Man Universal PCR Master Mix (Applied Biosystems) were used according to standard protocols. Beta-2 microglobulin (B2M Hs99999907_m1) (Applied Biosystems) was used for template normalization. RT-PCR was performed in three independent experiments, and duplicate determinations were carried out for each sample.

In our investigation we used nestin, β-tubulin III, GAP43 and doublecortin, because these genes and proteins play a key role in neurogenesis. 

Nestin refers to the VI type of intermediate filament proteins, it is more expressed in nerve cells, where it is responsible for axon growth in radial direction [44]. Nestin is expressed by various types of cells during their differentiation. One example of nestin expression is neural progenitor cells in the subgranular zone. The subgranular zone is a region of the brain located between a layer of granular cells and a chylus of the dentate gyrus of the hippocampus. In the subgranular zone postnatal neurogenesis occurs, that is the formation of new neurons from polypotent progenitor cells. Nestin is expressed in dividing cells at the early stages of their development in the central (CNS) and the peripheral nervous system [45]. After neuronal differentiation, nestin expression is suppressed and replaced by tissue-specific neurofilament proteins [46]. Thus, Nestin is used as a marker of progenitor cells in the CNS. 

GAP43 is a protein of neuronal plasticity, since high levels of its expression are observed in the cone of axon growth during its development, in axonal regeneration and after long-term potentiation [47]. This protein is a key component of the axon and presynaptic terminus. Mutation in the gene Gap43 leads to axon atrophy a few days after its formation [48]. Due to the cysteine site, GAP43 is able to bind to lipid rafts, the main components of cell membranes that coordinate neurotransmission and neuroplasticity [49,50].

β-tubulin III is a neuron-specific marker, which is associated with axonal compartment. This protein induces MSC differentiation into highly polarized neurons [51]. β-tubulin plays an important role in axonal transport. Mutations in β-tubulin gene are responsible for the molecular mechanism of human neuronal diseases [52].

Doublecortin is a microtubule-associated protein produced during neurogenesis. Doublecortin’s function is to stabilize microtubules and stimulates their polymerization, which allows migration of immature neurons to their designated location in the brain. Mutations in the doublecortin’s gene is the cause of severe brain formation disorders [53].

### 4.5. Statistical Analysis of the Experimental Data

Data were analyzed using GraphPad Prism version 6.0 software (GraphPad Software, La Jolla, CA, USA). The factor under investigation was mRNA expression. Data were expressed as a mean and standard deviation. A two-way analysis of variance tests was performed. Tukey tests were applied for pairwise comparisons. The value of *p* < 0.01 was considered statistically significant in all tests.

### 4.6. Molecular Modelling of AEDG Peptide—Histone Interactions

The calculation of spatial conformation of AEDG peptide was performed using the method of molecular mechanics (Figure 7).

This approach allows calculating the potential energy of a given system according to Hooke’s law: atoms in the molecule were considered as elastic balls of various sizes (according to the atom type) connected by springs of various lengths. The peptides were constructed in levorotatory conformation. In the calculations, total energy was minimized relative to the origin of coordinates:*E*_tot_ = *E*_str_ + *E*_bend_ + *E*_tors_ + *E*_vdw_ + *E*_elec_,(1)
where *E*_tot_ is the total potential energy of the macromolecule, *E*_str_ is the bond strain energy, *E*_bend_ is the valence angle strain energy, *E*_tors_ is the torsion angle strain energy, *E*_vdw_ is the Van der Waals interaction energy, and *E*_elec_ is the electrostatic interaction energy. The total steric energy of the system was calculated taking into account the force field, which contains a set of adjustable empirical parameters (force constants) and standard values of bond lengths, torsion, and valence angles. The Van der Waals interactions in the molecule were also considered. The first term of the equation describes the change in energy when the bond is stretched or compressed relative to its standard length:(2)$$E_{\mathrm{str}}=\frac{1}{2}k_{b}{\left(b-b_{0}\right)}^{2},$$
where *k_b_* is the constant force of bond stretching, *b*_0_ is the standard bond length, and *b* is the current bond length.

Angular deformations were described by the following equation:(3)$$E_{\mathrm{bend}}=\frac{1}{2}k_{0}{\left(\theta -\theta_{0}\right)}^{2},$$
where *k*_0_ is the constant force of valence angle deformation, *θ*_0_ is the equilibrium value of the valence angle, and *θ* is the current value of the valence angle.

The contribution of the rotation around dihedral angles to the potential energy was calculated using the following equation:(4)$$E_{\mathrm{tors}}=\frac{1}{2}k_{\phi }\left(1+\cos(n\varphi -\phi_{0}\right),$$
where *k*_ϕ_ is the torsion barrier (rotation barrier), *k_ϕ_* is the current value of the torsion angle, *n* is the period (the number of energy minima per one full cycle), and *ϕ*_0_ is the standard value of the torsion angle.

The van der Waals interactions between directly bound atoms are usually expressed through the Lennard‒Jones potential:(5)$$E_{\mathrm{vdw}}=\sum \frac{A_{ij}}{d_{ij}^{12}}-\frac{B_{ij}}{d_{ij}^{6}},$$
where *A_ij_* is the coefficient of the repulsion contribution, *B_ij_* is the coefficient of the attraction contribution, and *d_ij_* is the distance between atoms *i* and *j*.

Electrostatic forces are described by the function representing the expression for the Coulomb interaction:(6)$$E_{\mathrm{elec}}=\frac{1}{\varepsilon }\frac{Q_{1}Q_{2}}{d},$$
where ε is the permittivity, *Q*_1_ and *Q*_2_ are charges on the interacting atoms, and *d* is the interatomic distance.

When calculating molecule conformations, an important task was to correctly determine the force field in [54]. There are several ways to specify the force field: MMFF94x, Amber12EHT, BIO+, and OPLS, each differing by the degree of the possible approximations and assumptions. When using the MMFF94x field, potential fields formed by all atoms of the molecular system are considered. The method provides high accuracy and is suitable for small molecules. Amber12EHT is an all-atom force field developed to calculate conformations of proteins, nucleic acids, and small molecules in [55]. In this work, the conformations of short peptide molecules were calculated using the Amber12EHT force field. After the construction of peptide computer models, an important stage has been the geometrical optimization of their structures by minimizing energy using methods of steepest descent and conjugate gradients. The method of steepest descent is based on the gradual shift of atoms of the peptide molecule along the coordinate axes to find a new position with a lower potential energy. When the specified condition of energy minimum is reached, minimization stops. This method is used for structures that are far from the energy minimum. For a more accurate calculation, the conjugate gradient method is used, whose main idea is a gradual accumulation of information on the minimized function from iteration to iteration. The energy gradient is also considered at each stage of minimization, which is further used as additional information to calculate the new vector directions for the minimization procedure. Each subsequent stage continuously clarifies direction towards the minimum.

After geometric optimization, the molecule acquires many isomers, which correspond to different energy values. An important task is to find the most energy-efficient isomers of the molecule, which have similar values of total potential energy (*E*_tot_) within certain minima.

In our work, the conformational search was carried out using methods of molecular dynamics, whose purpose was to reproduce the motion of the molecule at a given time interval, in this case 1 fs. The basis of the method is the classical Newton equation of motion:(7)$$F_{i}\left(t\right)=m_{i}^{*}a_{i}\left(t\right),$$
where *F_i_*(*t*) is the force acting on atom *i* at time *t*, *m_i_* is the mass of the *i*-th atom, and *a_i_*(*t*) is the acceleration of atom *i* at time *t*.

At the final stage of the conformational search, the physicochemical features of the peptide conformers were evaluated, and the length of the peptide chain of the molecule, the charges, and the mean potential energy were calculated. Since peptides consisting of four amino acid residues were investigated in the study, only three torsion angles were measured. The number of conformations found in the peptides showed to what extent the molecules are spatially stable or unstable. It was assumed that the energetically most favorable stereoisomers of the molecules possess biological activity and perform functions in precisely this conformation.

To develop a computer model of the interaction of peptides with wheat histones, we designed their homologous models, since the spatial structures of wheat histones have not previously been resolved. To develop models of histones H1, H2b, H3, and H4, we used structures from the Protein Data Bank (PDB) 5NL0, Z chain; 1KX5, chain A and chain H; and 1EQZ, chain D.

Table 2 shows amino acid sequences of wheat histones and the corresponding histone sequences from the PDB database.

To construct histone models, we used the homologous modeling method, which allows estimating the degree of primary structure homology of the studied proteins using three-dimensional structures from the PDB database, resolved with X-ray structural analysis or nuclear magnetic resonance.

Active binding sites for peptides in histone molecules were found by the Edelsbrunner method in [56]. This method is based on the search for the most energetically preferable binding sites of peptides isolated in the histone structure by alpha-spheres. This method is used to determine regions of rigid atomic packing, but does not take into account sites that are too susceptible to solvent (sites located on the surface of the molecule). The sites were ranked by propensity of ligand binding (PLB), based on the amino acid composition of the pocket in [57]. Docking is a computer simulation of interactions between the ligand (peptide) and the active site of the receptor (histone). The docking method includes the placement of the ligand in different conformations at the binding site, as well as the calculation of the optimal mutual orientation of the peptide and histone molecules during their bind and the binding energy (kcal/mol). Semiflexible docking was used, where only the peptide conformational mobility was considered, while the side groups of the histones were rigid. When calculating the optimal spatial conformations of the interaction between peptides and histones, the contact area, the number of hydrogen bonds, as well as the parameters of hydrophobic and electrostatic interactions were considered. The Amber12EHT force field and the GBVI/WSA genetic search algorithm were used. Docking solutions were ranked according to the values of fitness function (∆G), which was calculated using the following formula:(8)$$\Delta G=c+\alpha \left(\frac{2}{3}\left(\Delta E_{\mathrm{coul}}+\Delta E_{\mathrm{sol}}\right)+\Delta E_{\mathrm{vdw}}+\beta \Delta SA_{\mathrm{weighted}}\right),$$
where *c* is the value of the loss of rotational and translational entropy of the complex; *α*, *β* are experimentally defined constants that depend on the force field; *E*_coul_ is the Coulomb energy value, which is calculated using the charge of the system with the permittivity of 1; *E*_sol_ is the value of the electrostatic energy of the solvent; *E*_vdw_ is the Van der Waals contribution to the interaction energy; *SA*_weighted_ is the contribution of molecular shells to the energy value.

Docking solutions were ranked in descending order from energetically most advantageous to least favorable. After analyzing the docking data, the energetically most favorable complex of peptide and histone was selected.

The ligand interaction module, which provides the opportunity to visualize the active site of the complex in the form of a diagram, is implemented in the Molecular Operating Environment 2016 programme. The amino acid residues in the active site are identified in two stages: first, receptor amino acids, ions, and solvent molecules that strongly interact with the ligand (interactions during formation of hydrogen bonds); second, receptor amino acids and ions that are close to the ligand, but with weak or diffuse interactions, such as collective hydrophobic or electrostatic ones.

## Figures and Tables

**Figure 1 molecules-25-00609-f001:**
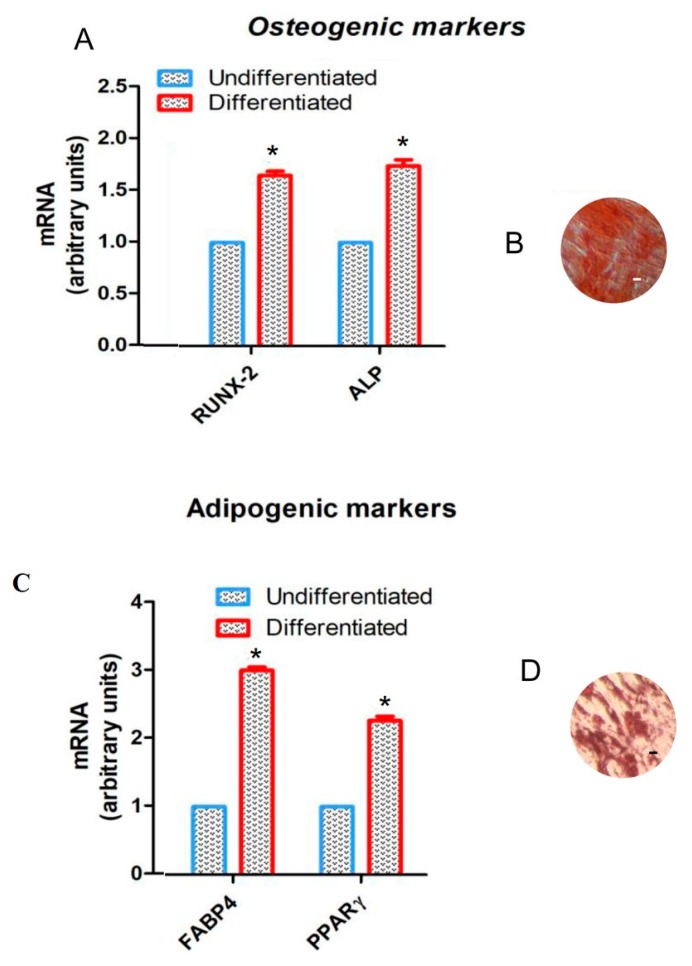
Cell characterization. (**A**) RT-PCR of osteogenic related markers, RUNX2 and ALP, performed in hGMSCs. Alizarin red S staining in (**B**) hGMSCs culture. (**C**) RT-PCR of adipogenic related markers, FABP4 and PPARγ, performed in hGMSCs. Oil red O staining in (**D**) hGMSCs culture. *, *p* < 0.01 statistically significant in comparison with the group “Undifferentiated”. Scale bar: 10 µm. Magnification: 10×.

**Figure 2 molecules-25-00609-f002:**
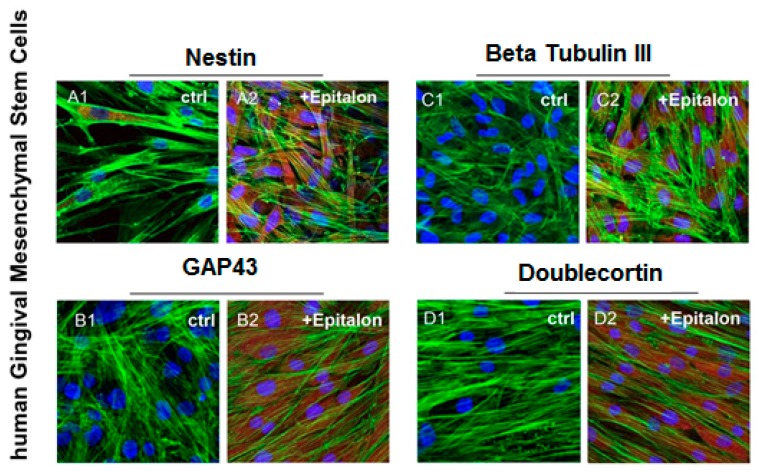
HGMSCs immunofluorescence analysis. Nestin expression in (**A1**) untreated hGMSCs and (**A2**) AEDG peptide (Ala-Glu-Asp-Gly, Epitalon) treated hGMSCs. GAP43 expression in (**B1**) untreated hGMSCs and (**B2**) AEDG peptide treated hGMSCs. Βeta Tubulin III expression in (**C1**) untreated hGMSCs and (**C2**) AEDG peptide treated hGMSCs. Doublecortin expression in (**D1**) untreated hGMSCs and (**D2**) AEDG peptide treated hGMSCs. Scale bar: 10 µm. Magnification: 20×.

**Figure 3 molecules-25-00609-f003:**
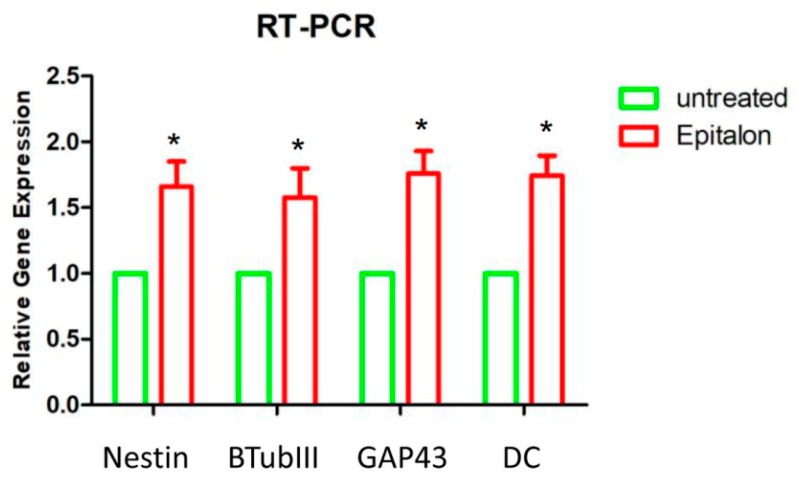
RT-PCR. Gene expression of neurogenic related markers in hGMSCs treated with AEDG peptide (Ala-Glu-Asp-Gly, Epitalon). *, *p* < 0.01 statistically significant in comparison with the group “untreated”.

**Figure 4 molecules-25-00609-f004:**
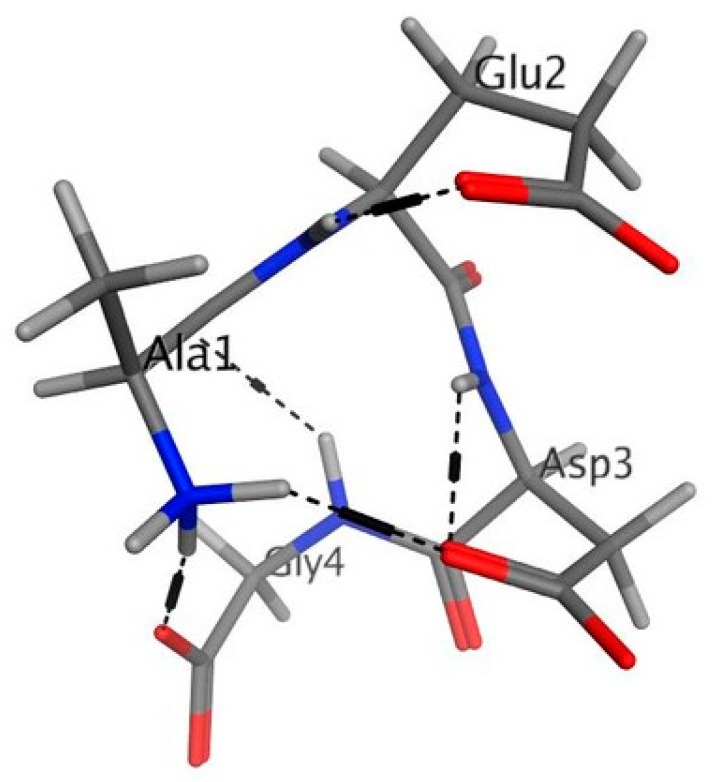
Low energy conformation of the AEDG peptide (Ala-Glu-Asp-Gly, Epitalon). Oxygen atoms are red, nitrogen atoms are blue, carbon atoms are black, hydrogen atoms are light gray, and hydrogen bonds are dotted.

**Figure 5 molecules-25-00609-f005:**
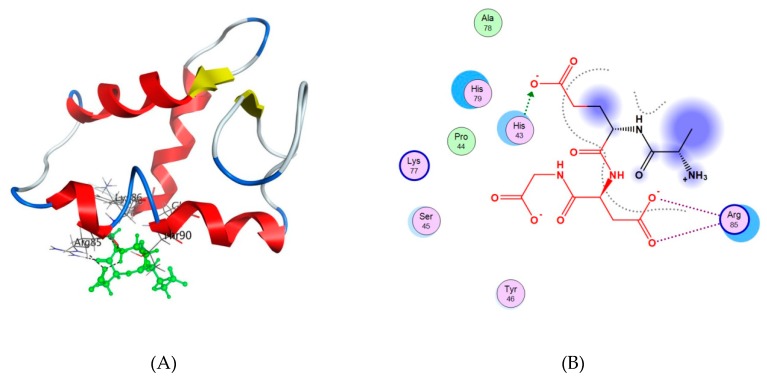
(**A**) The interaction of the AEDG peptide (Ala-Glu-Asp-Gly, Epitalon) with histone H1/6. The histone molecule is shown as α-helical domains and loops. Oxygen atoms are red, nitrogen atoms are blue, carbon atoms are black, and hydrogen atoms are light gray. The peptide is green. The dotted line shows hydrogen bonds; 6 (**B**) Scheme of interaction of the AEDG peptide with histone H1/6 at site 5: Tyr46-Arg85-Lys86-Thr90-Gln91. The arrows show the direction of proton transfer in the donor—acceptor pair. The gray dotted line shows the ligand region accessible to the solvent. The blue circles depict amino acids that are close to the receptor molecule but do not interact with it; however, they can have an effect on the orientation and binding of the molecule.

**Figure 6 molecules-25-00609-f006:**
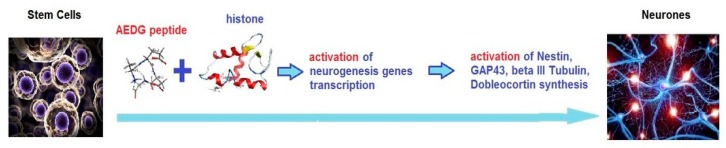
Possible scheme of peptide epigenetic regulation of stem cells neuronal differentiation.

**Figure 7 molecules-25-00609-f007:**
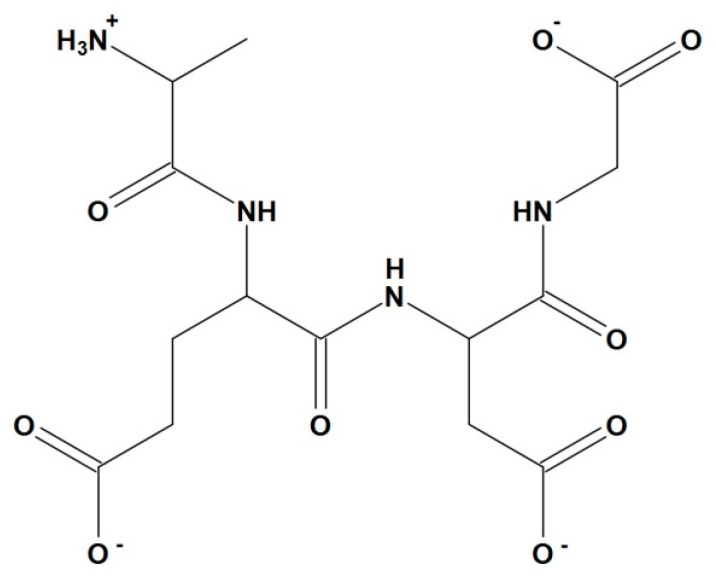
2D-structure of AEDG peptide (Ala-Glu-Asp-Gly, Epitalon) in protonated form with pH 7.

**Table 1 molecules-25-00609-t001:** Cytofluorimetric analysis.

Antigen	hGMSCs	
	Phenotype	MFI Ratio ± SD
CD13	+++	173.4 ± 21.7
CD29	+++	195.2 ± 22.3
CD44	+++	167.8 ± 27.3
CD73	++	29.4 ± 2.9
CD90	+++	396.4 ± 28.2
CD105	+	9.1 ± 1.3
CD14	−	ND
CD34	−	ND
CD45	−	ND

Note: − indicates negative expression (0%); + indicates moderate expression; ++ indicates positive; +++ indicates high expression (100%); MFI ratio is the average of three different biological samples ± standard deviation.

**Table 2 molecules-25-00609-t002:** Amino acid sequences of histones.

Histone of Wheat *Triticum aestivum*, FASTA	PDB Structure and Its Amino Acid Sequence, Used to Construct Homologous Histone Models, FASTA	Identity of Original Sequence and Template, %
>H1.1 MSTDVVADVPAPEVAAAADPVVETTAEPAAGDANAAKETKAKAAKAKKPSAPRKPRAAPA**HPTYAEMVSEAITALKERTGSSPYAIAKFVEDKHKAHLPANFRKILSVQLKKLVASGKLTKVKASYKLSAAAAKPK**PAAKKKPAAKKKAPAKKTATKTKAKAPAKKSAAKPKAKAPAKTKAAAKPKAAAKPKAKAPAKTKAAAKPKAAAKPKGPPAKAAKTSAKDAPGKNAGAAAPKKPAARKPPTKRSTPVKKAAPAKKAAPAKKAPAAKKAKK	>5NL0_Z **HPKYSDMILAAVQAEKSRSGSSRQSIQKYIKNHYKVGENADSQIKLSIKRLVTSGALKQTKGVGASGSFRLAK**	34
>H1.3 MSTEVAAADIPVPQVEVAADAAVDTPAANAKAPKAAKAKKSTGPKKPRVTPA**HPSYAEMVSEAIAALKERSGSSTIAIGKFIEDKHKAHLPANFRKILLTQIKKLVAAGKLTKVKGSYKLAKAPAAV**KPKTATKKKPAAKPKAKAPAKKTAAKSPAKKAAAKPKAKAPAKAKAVAKPKAAAKPKAAAKPKAKAAAKKAPAAATPKKPAARKPPTKRATPVKKAAPAKKPAAKKAKK	>5NL0_Z **HPKYSDMILAAVQAEKSRSGSSRQSIQKYIKNHYKVGENADSQIKLSIKRLVTSGALKQTKGVGASGSFRLAK**	36
>H1.6 PVPQVEVAADAAVDTPAASAKAPKAAKAKKSTGPKKPRVTPA**HPSYAEMVSEAIAALKERSGSSTIAIAKFIEDKHKAHLPANFRKILLTQIKKLVAAGKLTKVKGSYKLAKAPAAV**KPKTATKKKPAAKPKAKAPAKKTAAKSPAKKAAAKPKAKAPAKAKAVAKPKAASKPKAAAKPKAKAAAKKAPAAATPKKPAAARKPPTKRATPVKKAAPAKKPAAKKAKK	>5NL0_Z **HPKYSDMILAAVQAEKSRSGSSRQSIQKYIKNHYKVGENADSQIKLSIKRLVTSGALKQTKGVGASGSFRLAK**	36
>H2b MAPKAAKKPAAKKPAEEEPAAEK**AEKTPAGKKPKAERRIPAGKSAAKAGGDKKGKKKAKKSVETYKKYIFKVIKQVHPDIGISSKAMSIMNSFINDIFEKLAGEAAKIARYNKKPYITSREIQTSVRLVLPGELAKHAVSEGTKAVTKFT**SAS	>1KX5_H **PEPAKSAPAPKKGSKKAVTKTQKKDGKKRRKTRKESYAIYVYKVLKQVHPDTGISSKAMSIMNSFVNDVFERIAGEASRLAHYNKRSTITSREIQTAVRLLLPGELAKHAVSEGTKAVTKYTSAK**	65
>H3 **ARTKQTARKSTGGKAPRKQLATKAARKSAPATGGVKKPHRERPGTVALREIRKYQKSTELLIRKLPFQRLVREIAQDFKTDLRFQSSAVAALQEAAEAYLVGLFEDTNLCAIHAKRVTIMPKDIQLARRIRGERA**	>1KX5_A **ARTKQTARKSTGGKAPRKQLATKAARKSAPATGGVKKPHRYRPGTVALREIRRYQKSTELLIRKLPFQRLVREIAQDFKTDLRFQSSAVMALQEASEAYLVALFEDTNLCAIHAKRVTIMPKDIQLARRIRGERA**	96
>H4 SGRGKGGKGLGKG**GAKRHRKVLRDNIQGITKPAIRRLARRGGVKRISGLIYEETRGVLKIFLENVIRDAVTYTEHARRKTVTANDVVYALKRQGRTLYGFGG**	>1EQZ_D **GAKRHRKVLRDNIQGITKPAIRRLARRGGVKRISGLIYEETRGVLKVFLENVIRDAVTYTEHAKRKTVTAMDVVYALKRQGRTLYGFGG**	97

Note: Amino acid sequences of histones with resolved secondary structure taken from PDB are highlighted in bold. The identity of the sequences was calculated using BLASTp (https://blast.ncbi.nlm.nih.gov).

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
