# Peer review of "AEDG Peptide (Epitalon) Stimulates Gene Expression and Protein Synthesis during Neurogenesis: Possible Epigenetic Mechanism"

_molecules, 2020, doi:10.3390/molecules25030609_

Round 1
Reviewer 1 Report
This is a pretty interesting study that evaluates a possible mechanism of action of the AEDG peptide (Epitalon) during the known effect on neuronal differentiation of human stem cells. The ability of Epitalon, in comparison with other peptides, to induce neuronal differentiation of stem cells of periodontal origin has been recently already explored by these authors (International journal of immunopathology and pharmacology 2019, doi:10.1177/2058738419828613), which now examined the same effect on human gingival mesenchymal cell, and hypothesized an epigenetic nature of the mechanism of gene expression modulation.
Overall, this is a well written manuscript, the results presented clearly, but I have concerns about originality and the conclusions appear rather speculative. The claimed epigenetic mechanism is only hypothesized on the basis of calculated histone-peptide interaction obtained by molecular modeling analysis. The epigenetic mechanisms have a unique impact on vertebrate development and occur as DNA methylation, histone posttranscriptional modification (acetylation/deacetylation), and non-coding RNAs. The study will reach, therefore, a more completeness by probing this mechanism of histone mediated modulation of gene expression as with FITC-Labeled Histones in Fedoreyeva et al. 2013, and further exploration by more specific experiments (the use of negative control peptide with scramble or reverse sequences) to strengthen their conclusions/findings and the novelty of this otherwise interesting study.
I have also a minor point to raise:
Lane 517: Fedoreeva is misspelled; should be Federeyeva.
Author Response
Dear reviewer, please, see the attachement.
Thank you.
Best regards,
Natalia Linkova

Reviewer 2 Report
This is a good manuscript with good data quality and clear presentation. The results are convincing. While the English writing and labeling need to be improved.
My minor concern is:
Beta tubulin III is marked as “B tubulin III” in the figure.
Author Response

(The authors gave the same response as above.)

Reviewer 3 Report
Comments on Manuscript entitle: “AEDG Peptide (Epitalon) Stimulates Gene 2 Expression and Protein Synthesis during 3 Neurogenesis: Possible Epigenetic Mechanism” by Vladimir Khavinson et al.
This manuscript is interesting and well enough written, however there are still some questions to be addressed, and tasks to improve concerning the manuscript:
1) Line 66: are all short peptides able to interact with DNA and histone proteins. The authors should better indicate the characteristics of these particular peptides or give examples.
2) Figure 2: Why is the standard deviation not indicated on the histograms for undifferentiated cells?
3) Line 109: the authors should indicate the concentration of AEDG used for the treatment
4) Gene expression:
- Why is the standard deviation not indicated on the histograms for untreated cells?
- the authors should at least use a negative control or a scramble peptide to emphasize the importance of the AEDG structure in the upregolation of the genes in question.
5) Line 184-186: the authors should indicate at least the number, gender and age of the volunteers from whom the cells were taken
6) the authors should indicate the concentration expressed as molarity, or even as molarity
Author Response

(The authors gave the same response as above.)

Round 2
Reviewer 1 Report
Thank you for your answer. Good luck.
Reviewer 3 Report
The authors responded adequately to the questions submitted. I do not think further modifications are necessary and for this reason I consider this manuscript suitable for publication.